# Physical Education with Eduball Stimulates Non-Native Language Learning in Primary School Students

**DOI:** 10.3390/ijerph19138192

**Published:** 2022-07-04

**Authors:** Ireneusz Cichy, Agnieszka Kruszwicka, Patrycja Palus, Tomasz Przybyla, Rainer Schliermann, Sara Wawrzyniak, Michal Klichowski, Andrzej Rokita

**Affiliations:** 1Department of Team Sports Games, Wroclaw University of Health and Sport Sciences, Mickiewicza 58, 51-684 Wroclaw, Poland; paluspatrycjaa@gmail.com (P.P.); sara.wawrzyniak@awf.wroc.pl (S.W.); andrzej.rokita@awf.wroc.pl (A.R.); 2Learning Laboratory, Adam Mickiewicz University, Szamarzewskiego 89, 60-568 Poznan, Poland; a.kruszwicka@gmail.com (A.K.); tomekprzybyla@gmail.com (T.P.); 3Faculty Social and Health Care Sciences, Regensburg University of Applied Sciences, Seybothstraße 2, 93053 Regensburg, Germany; rainer.schliermann@oth-regensburg.de

**Keywords:** educational balls, dual-language schools, gross motor, learning, locomotor skills, object control, primary education, second language skills

## Abstract

Although the neuronal mechanisms of action and cognition are related, the division of intellectual and physical lessons is standard in schools. This is surprising, because numerous studies show that integrating physical education (PE) with teaching content stimulates critical skills. For example, several experiments indicate that Eduball-based PE (i.e., lessons in a sports hall during which students play team mini-games with educational balls with printed letters, numbers, and other signs) develops mathematical and language competencies. At the same time, the Eduball method does not slow down learners’ physical development. However, we have little knowledge about the effects of such techniques on non-native language learning. Consequently, the absence of incorporating core academic subjects into PE in dual-language schools or during foreign language education is exceptionally high. Here, we replicated the Eduball experiment, but with the goal of testing this method for non-native language learning. Thus, the intervention occurred in a dual-language primary school and we evaluated second language (L2) learning. As before, we used the technique of parallel groups (experimental and control); in both groups, there were three 45-min PE classes per week. In the experimental class, two of them were held using Eduball. After a half-year experiment, children from the experimental group (one second-grade, *N* = 14) improved their non-native language skills significantly more than their peers from the control group (one second-grade, *N* = 12). These findings demonstrate that Eduball-type intervention stimulates non-native language learning in children. Hence, our report suggests that specific body training forms can support L2 learning.

## 1. Introduction

Integrating physical education (PE) with teaching content benefits children’s physical, social, and intellectual development [1,2,3,4]. This approach mainly stimulates critical school skills such as reading, writing, and mathematical competencies [5,6,7,8,9,10,11,12,13,14,15,16]. However, despite considerable evidence and the growing interest in implementing such a strategy in early childhood education, the traditional division into cognitive and physical lessons is still widely practiced in schools [13,17,18]. This is primarily because the curriculum is crowded and PE is perceived as the least essential school subject [13]. Moreover, methods based on simultaneous motor-cognitive tasks are still lacking [17]. Consequently, children are not physically active enough at school. They also learn only through selected modalities, marginalizing the crucial role of bodily experiences in the learning process [19,20,21]. Therefore, the Eduball method was created.

Eduball enables merging movement with learning content using a set of educational balls for team mini-games printed with letters, numbers, and other signs. Our studies to date [22,23,24,25,26,27,28] show that participation in the Eduball classes, or in classes with very similar balls called SmartBall, used by other authors [29], influences both motor and academic achievements. For the latter, the Eduball method improves various aspects of fundamental school skills, such as mathematical and language dexterities [24,25,26]. In the case of language, Eduball stimulates, for example, the development of the ability to write straight within lines, as well as reading performance [25]. Notably, the Eduball method is easy-to-use and does not require a drastic change in the curriculum or any special methodical preparation. Furthermore, all types of child teachers (i.e., regular school teachers, PE teachers, and both of them in collaboration) can use Eduball effectively to develop cognitive and motor skills in school students [28]. Additionally, Eduball can be successfully used as therapeutic support for low-performing students, particularly children diagnosed with such complex language disorders as dyslexia. In this case, the Eduball method contributes, for example, to equalizing learners’ educational opportunities [27]. Several other (non-Eduball) studies (e.g., [30,31]) also indicated that ball-based interventions (e.g., TherapyBall or SensoryBall) stimulate the linguistic development of children with dyslexia and other developmental disabilities.

Despite such clear evidence that approaches integrating PE with teaching content (e.g., Eduball) positively influence the acquisition of language skills, there is a knowledge gap concerning their effects on children’s skills in languages other than their native tongue. We know that students perceive bilingual PE as an attractive form of class [32], but its effectiveness in learning a non-native language has not been investigated yet. We also know that people who train intensively in gross motor skills, e.g., athletes, learn English as a foreign language (EFL) faster [33], but we do not know whether it is the effect of physical activity or travel and meetings with other players during which English is used. There are some suggestions from research on embodied cognition that merging non-native language learning with fine motor actions (e.g., with gestures) leads to improved language achievements (e.g., in preschool children) [34]. This is explained by the fact that motor and language functional networks in the brain are closely related [35,36,37,38,39,40,41]. Moreover, first (L1) and second language (L2) representations are co-lateralized (typically to the left hemisphere, exactly like gestures) [35,38]. Nevertheless, despite this co-lateralization, the L1 and L2 control/neural mechanisms are different, and L1 and L2 have no identical cortical organization [38]. In other words, native and non-native languages are processed in close, but distinctive cortical circuits [42,43]. Thus, their connections with motor networks are also varied [37,38]. Therefore, we do not know whether or not the impact of integrating PE with teaching content on non-native language learning is similar to that of native language.

To address this knowledge gap, we replicated our previous Eduball experiment with the participation of Polish-speaking students. In short, the intervention was that various Eduball games were incorporated into PE lesson plans in a primary school for half a year. However, as our goal was to test the Eduball method for non-native language learning, the experiment occurred in a dual-language (Polish–English) school. Moreover, we used the English test to measure second language learning instead of cognitive tests evaluating crucial school skills. We assumed that although L1 and L2 do not have identical neuronal representations, both languages’ learning process is strongly embodied. Thus, we hypothesized that an Eduball-type intervention should stimulate non-native language learning in children.

## 2. Materials and Methods

### 2.1. Participants

Twenty-six Polish students from two second-grade classes (12 girls, age: 7–8, mean = 7.54, *SD* = 0.51) participated in the experiment. Both classes attended the same dual-language (Polish–English) school located in a large city in Poland. Classes were randomly assigned to control (C) and experimental (E) groups. The control group included 12 students (5 girls, mean age = 8.00, *SD* = 0.00), while the experimental group comprised 14 students (7 girls, mean age = 7.14, *SD* = 0.36). Randomization was performed using Research Randomizer—a random group generator available at https://www.randomizer.org/ (accessed on 28 August 2017). Inclusion criteria were as follows: being a student of the selected class and regularly participating in all activities during the experimental period. Exclusion criteria were as follows: contraindications for participation in PE and missing pre- or/and post-test. All students met the inclusion criteria and none were subsequently excluded from the experiment.

### 2.2. Experimental Factor

The experimental factor was PE carried out using Eduball. The Eduball set consists of 100 educational balls used for team mini-games. The balls are divided into two main subgroups. The first contains balls 57 cm in circumference (close to size 3 basketballs) and 310 g of weight. These balls are in yellow and green colors (40 in each color) and are printed on one side with uppercase and the opposite side with lowercase black letters and numbers from 0 to 9 (the same on the top and bottom side) (see Figure 1a). The second subgroup includes balls 63 cm in circumference (close to size 4 volleyballs) and 250 g of weight. These balls are in red and blue colors (eight in each color). On their surfaces, mathematical symbols such as addition (+), subtraction (−), multiplication (*), division (:), greater than (>), less than (<), and (parentheses), as well as the ‘at’ sign (@), are painted. Additionally, this set contains four unprinted orange balls that could be used as universal blanks (see Figure 1b).

Precisely as in the Eduball experiments replicated here (for more details, see [24,25,26,27,28]), various activities from the Eduball games set [44,45] were incorporated into PE lesson plans. However, the games were selected not only for the thematic cycle and the day’s theme (in compliance with the curricula) as in the past Eduball studies, but also for the current level of development in a non-native language. Therefore, while students practiced in class, for instance, recognizing and writing the letters of the English alphabet, the Eduball games featured exercises related to identifying the letters of the alphabet. Nevertheless, all the activities were based on the core curriculum. Most of all, it is worth emphasizing that adapting the games to the students’ English language level never changed the idea of Eduball, and each game reflected the basic principles of the method (for detailed descriptions of the Eduball method, see [22,23,24,25,26,27,28]).

### 2.3. Procedure

Our investigation protocol was assessed and approved by the local Ethics Committee for Research Involving Human Subjects (Resolution #37/2016 of the Senate Committee on Ethics of Scientific Research at the Wroclaw University of Health and Sport Sciences on 16 October 2016). Furthermore, informed consent was obtained from all participants’ parents or legal guardians involved in the experiment. Finally, the experiment was conducted in accordance with the principles of the Helsinki Declaration.

The experiment lasted five months and was performed during the first semester (which in Poland, in the school year 2017/2018, begins in September and ends in January) in natural conditions (at school) using the technique of parallel groups (experimental and control). In both groups, there were three 45 min PE classes per week. The PE teacher taught all PE classes. In the experimental class, two 45 min PE classes a week were held using Eduball. In the control class, all PE classes were conducted without Eduball, following the standard PE program. In other words, in the control group, the PE teacher conducted the PE program under the aims and objectives of the school’s program for developing physical fitness and health education.

The experimental group and the control group followed the same curriculum based on the core program of the Polish National Ministry of Education. However, since it was a dual-language (Polish-English) school, the curriculum was implemented not only in the native language (Polish), but partially in the non-native language (English). Therefore, PE in the control group was—similarly to the experimental group—partially conducted in the non-native language (the rule was that only complex instructions were given in Polish, so the teacher formulated simple communications in English and, if possible, also conversed with students in this non-native language).

Our study involved two measurement periods: a pre-test at the beginning of the school year (September) and a post-test at the end of the first semester (January). During both of them, fundamental motor skills were diagnosed using the Test of Gross Motor Development (Second Edition) and foreign language skills by applying the Pre A1 Starters (the structure of our experimental workflow is depicted in Figure 2). Specially trained researchers administered the pre- and post-test.

#### 2.3.1. Test of Gross Motor Development—Second Edition (TGMD-2)

We evaluated movement skills using the Test of Gross Motor Development—Second Edition (TGMD-2). It is a reliable and valid test applied worldwide, designed to examine locomotor (run, gallop, hop, leap, jump, and slide) and object control (strike, dribble, catch, kick, throw, and roll) skills [46]. We conducted this test exactly as described in our earlier Eduball works [26,28]. In short, the TGMD-2 testing was conducted during a school PE class at the sports hall by four experimenters (one supervisor and three students of the Wroclaw University of Health and Sport Sciences). First, one experimenter demonstrated the proper execution of locomotor and object control abilities. Next, the children re-demonstrated these actions in the same order. The learner had to complete one practice and then two formal trials. During this time, all experimenters observed and scored each re-demonstration for every trial on the spot based on three to five performance criteria (e.g., for a run: arms moved in opposition to legs, elbows bent; a brief period where both feet were off the ground; narrow foot placement landing on the heel or too; non-support leg bent approximately 90 degrees; for more examples see [46,47]). Experimenters scored qualitative performance criteria to evaluate the skill performance. They used the binary 0–1 scale (1 = the presence of a performance criterion for given motor skill, 0 = the absence of the performance criterion). The highest total score for the locomotor and object control skills was 48 points (24 per each subtest; the higher the total score, the better the performance). For administering the TGMD-2, we used: one ~9-inch playground ball, one basketball, one soccer ball, one 4-inch lightweight ball, one tennis ball, one softball, one ~4.5-inch square beanbag, tape, two traffic cones, one plastic bat, and one batting tee [46].

#### 2.3.2. Cambridge English: Starters (Pre A1 Starters)

To test non-native language skills, we used the Pre A1 Starters test, also known as Cambridge English: Starters (YLE Starters). It is one of the Cambridge English Qualifications in-depth exams designed for young learners. Pre A1 Starters is embedded in the European Language Education System (CEFR) and is commonly used worldwide. This test comprises three parts: (1) listening, (2) reading and writing, and (3) speaking, where each category has a different number of exercises. For example, listening consists of four parts with 20 questions and lasts about 20 min (activities are designed to show how well a student can understand simple English words and grammar—children tick the correct picture to show they have understood a conversation). Reading and writing comprised five parts with 25 questions and lasted about 20 min (students show they can read and understand simple text in English—they write sentences to describe the picture). The last category (speaking) lasts about 3 to 5 min and consists of four parts (children describe the difference between two pictures). All parts last about 45 min. The test format could be computer-based or paper-based. The full Pre A1 Starters description is available at https://www.cambridgeenglish.org (accessed on 16 May 2022).

We ran the Pre A1 Starters in the students’ classroom. During the test, the participant and the experimenter were together in this room. As the test format was paper-based, we adjusted all infrastructure elements of the room (e.g., chairs and tables) to the age of the pupils. When the test started, the researcher introduced their aims. During this time, the participants were also familiarized with how to fill it. For instance, the experimenter explained that the reading and writing tasks would be performed with only an ordinary pen. In contrast, the listening tasks would be performed with an ordinary pen and additional colored pens (red, blue, green, yellow, orange, pink, purple, black, brown, and grey). The researcher also emphasized that in the speaking part, she or he will repeat instructions two times, and in listening, the student will hear each recording twice. For each component, the student could receive a maximum of five shields. Five shields mean that the child did very well in the given part. The number of shields for all parts together was expressed as a percentage—the greater the percent, the better the learner’s performance.

### 2.4. Data Analysis

The primary dependent variables were represented by non-native language skills and fundamental movement (locomotor and object control) skills. These variables were expressed in the mean scores and calculated separately for the control and experimental groups and pre-test and post-test. Since we performed precisely the same data analysis as in earlier Eduball studies [26,28], we only describe it briefly here. Initially, we applied the paired samples *t*-test to compare the changes in the score (pre-test vs. post-test) within the control and experimental groups. Next, we ran an analysis of covariance (*ANCOVA*) to determine the significant difference between groups (control vs. experimental) after the experiment. We set learners’ pre-test scores as the covariate, with post-test scores as the dependent variable. Additionally, we ran a one-way *ANOVA* with group type (C vs. E) as a factor to confirm that there were no significant differences between the groups in the pre-test. The adopted level of significance was *α* = 0.05. The statistical analyses were carried out using *jamovi* for Mac (Version 2.3) [48,49,50]. However, the post-hoc power analysis was conducted using G*Power (Version 3.1.9.7). All anonymized data supporting this study’s findings are publicly available in the Open Science Framework at https://osf.io/q8scy/ (accessed on 8 June 2022).

## 3. Results

To confirm that there were no significant differences between the groups at the beginning of the experiment, pre-test values for non-native language, locomotor, and object control skills were compared with a one-way *ANOVA* with group type (C vs. E) as a factor. There was only a main effect for object control skills (*p* < 0.01), such that C scored higher than E (see Table 1). No other differences between C and E were found (all the remaining *p* > 0.20, see Table 1).

As depicted in Table 2, a *t*-test-based comparison of the pre-test scores with the post-test scores showed that both groups significantly improved their non-native language skills after half a year of our experiment. However, a one-way *ANCOVA* (see Table 3) indicated a significant difference in this improvement, as the Eduball group made more progress than the non-Eduball one. These results are expressed as estimated marginal means in Figure 3a. Note that the post-hoc power analysis revealed that for the observed (large, according to Cohen’s criteria) effect size (*f* = 0.90), a calculated power is 0.99 (when the required level is 0.80), which indicates that the sample size was large enough for the analysis conducted here.

We did not observe any deterioration in gross motor development in the experimental group. While we revealed a slight weakening in dribbling, a similar trend was found in the control group (*p* < 0.09), where we detected an overall deterioration in object control skills (see Table 2). Therefore, the groups at the end of the experiment did not vary in terms of locomotor development. However, the classes differed in object control development, as it worsened in the non-Eduball group (see Table 3). These findings are expressed as estimated marginal means in Figure 3b (locomotor skills data) and Figure 3c (object control skills data).

## 4. Discussion

Replicating the Eduball experiment in a new, dual-language context, we confirmed the hypothesis that integrating PE with a non-native language stimulates L2 learning in children. Moreover, we have found that such an intervention does not slow down students’ motor development. Even though our study was short (it lasted only half a year/one semester), we detected that the students from the Eduball group improved their non-native language competencies significantly more than their peers from the non-Eduball group. Notably, at the end of the experiment, both groups represented similar motor levels (in some aspects, the control group was even lower). Therefore, our study corroborates that specific (and well thought out) forms of cognitive-motor integration can be an effective and safe strategy for supporting L2 learning. In other words, psycho-physiological changes in response to such exercises may have similar implications for non-native language functions as in native ones [52,53]. Moreover, this study suggests that acquiring competencies in L2 is similarly embodied, as is the case in L1. However, future investigations are needed to better understand the mechanisms that interface motor control and non-native language representations.

### 4.1. Eduball and Non-Native Language Learning

The fact that we have revealed that Eduball positively affects children’s native [25,27,28] and non-native language skills does not entitle us to generalize the results to all cases. Although we know from previous studies [35,37,38] that for Polish (as L1) and English (as L2) the neuronal control mechanisms are somewhat different and often independent (this also applies to other language pairs, such as Macedonian–English [43]), these two languages are similar when we analyze them in the context of embodied linguistic cognition. For example, both have egocentric points of view, i.e., internal position simulations have a reference point in the body [54]. This is different from languages with a geocentric system—here, the references refer to points in space, distance, or directions of the world [55]. Hence, while learning geocentric languages is still underpinned by the brain’s sensory-motor mechanisms [56,57], which is of great importance for the assimilation and representation of conceptual knowledge [56,58], it should be less embodied compared to the egocentric case [59,60,61,62]. Therefore, future research should take into account pairs of differently represented languages. It would be fascinating to intervene with native speakers of a geocentric language, such as Kuuk Thaayorre (a Paman language spoken in Pormpuraaw in Australia by the Thaayorre people) [63], who are learning an egocentric language, such as English (EFL).

Additionally, in the case of EFL, children from countries with a different share of English in their everyday life and popular culture should be examined. This is because, for instance, when TV series or cartoons are often shared with original English-speaking dialogues in a given country, the development of vocabulary knowledge and speaking skills by young English learners is stimulated [64,65]. This is the case, for example, in China, where movies, series, cartoons, TV shows, and TV commercials imported from English-speaking countries are the standards. Moreover, in this country, students often consciously watch English videos to learn [66]. This is similar in Poland—children and youth are eager to watch English-language series to improve their non-native language [67]. Nevertheless, there are countries where such interest is not so great (e.g., Germany) [67]. Such cultural determinants of L1 and L2 learning should also be considered in future experiments.

### 4.2. Eduball and Motor Development

Our outcomes regarding motor skills align with previous observations that when using Eduball, there is no risk of deterioration in physical fitness [24,25,26,27,28]. In other words, we again confirmed that incorporating additional academic instructions into PE (native language, mathematics, or, as in this study, English content) has no adverse effects on motor performance. However, in this project, we did not observe that the Eduball intervention stimulated motor development, as in some earlier Eduball works (e.g., [26,28]). This is a surprise because innovations introduced to PE most often activate children, and intensified physical development has been noticed [68,69,70]. Nevertheless, both Eduball studies where increases have been detected (e.g., [28]) and non-Eduball studies that we mentioned were based on interventions longer than half a year (e.g., one year), or a post-test was completed several months after the end of the intervention (measuring long-term effects, e.g., [26]). In other typical one-semester Eduball experiments (e.g., [25]), such additional motor effects did not occur. Thus, the demonstrated lack of improvement in motor skills is probably related to the too-short influence time (only half a year) or too-early motor testing. This issue should be addressed in future research more focused on the physiological effects of Eduball and on other motor skills, such as space-time orientation or graphomotor dexterities (which are based mainly on the praxis network closely related to linguistic or mathematical cognition [71,72,73,74]). Note that in this report, we concentrated on non-native language skills and used TGMD-2 only to check that the experimental factor was not impairing physical development.

We must emphasize that what distinguishes Eduball-based PE from traditional PE is not only the addition of cognitive elements. Physical activity itself is a bit different here. The essential (motor) assumption of the Eduball methods refers to the observation that although the human body has a symmetrical appearance from the external point of view, it is functionally asymmetrical [75]. However, most activities in traditional PE, such as handball, tennis, or high jumping, are dominated by unilateral motor practice [76] and increase functional asymmetry even more [77]. The opposite is the case with Eduball-based PE, which strives for symmetry [27]. To reduce asymmetry and to develop the body and the brain holistically, the Eduball method, in a way, forces students to practice bilateral [78] and non-dominant hand (as well as leg) training [79,80]. Such an approach is focused on intensive cognitive development while stimulating physical development [27]. Nonetheless, it may also contribute—as the effect of an interhemispheric transfer—to increasing the effectiveness of PE in the context of physical parameters, especially in terms of object control skills [81,82]. Perhaps if the control group used Eduball, there would be no deterioration of these competencies, as was the case in the experimental/Eduball group. Therefore, we reiterate that there is a clear need for research focused on the physiological effects of Eduball. However, it seems we can point out some practical implications. In the Eduball method or similar methods (e.g., SmartBall), the teacher can focus on any school content without exposing students to any risk in motor development.

### 4.3. Limitations

One of the potential limitations is the use of only one pair of languages (Polish as native and English as non-native). Thus, future research should take into account pairs of other languages. Moreover, EFL learning in a country such as Poland (an EU member) is quite intuitive, as many of the elements of popular culture here have English-speaking roots. Therefore, students from other cultural conditions of EFL learning should be involved in upcoming experiments. Finally, to better assess the impact of the tested method on motor development, the next project should be extended for another semester and/or follow-up tests should be applied (e.g., after six months). Additionally, physiological and other motor-skill (such as space-time orientation or graphomotor dexterities) tests should be used. Future investigations could also include an additional control group in which a different PE innovation would be applied to determine whether the observed effects are related to the tested method or the innovation *per se*.

## 5. Conclusions

Previous studies demonstrate that the Eduball method, i.e., an approach integrating PE with teaching content, positively influences the acquisition of crucial school skills such as mathematical and native language abilities. Here, we show that this strategy also stimulates non-native language learning in children. After a half-year experiment, students from the experimental (Eduball) group improved their non-native language skills significantly more than learners from the control group who participated in traditional PE. Moreover, we found that adding foreign language content to PE does not adversely affect the physical development of students. Therefore, our results suggest that L2 learning, as in the case of L1, is firmly embedded and can be stimulated by specific forms of motor-cognitive training. Hence, our outcomes, although with some limitations, fit into the broad debate on embodied linguistic cognition, but refer to the little-studied context of a non-native language.

## Figures and Tables

**Figure 1 ijerph-19-08192-f001:**
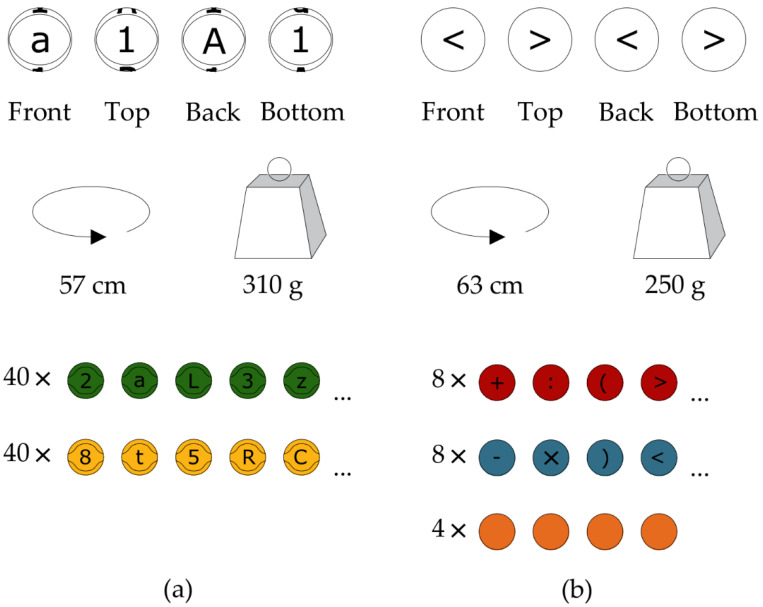
Eduball educational balls. (**a**) The first type of balls are in yellow and green colors (40 in each color) and are printed on one side with uppercase and the opposite side with lowercase black letters and numbers from 0 to 9. (**b**) The second type of balls are in red and blue colors (eight in each color), and mathematical symbols are painted on their surfaces; this set also contains unprinted orange balls.

**Figure 2 ijerph-19-08192-f002:**
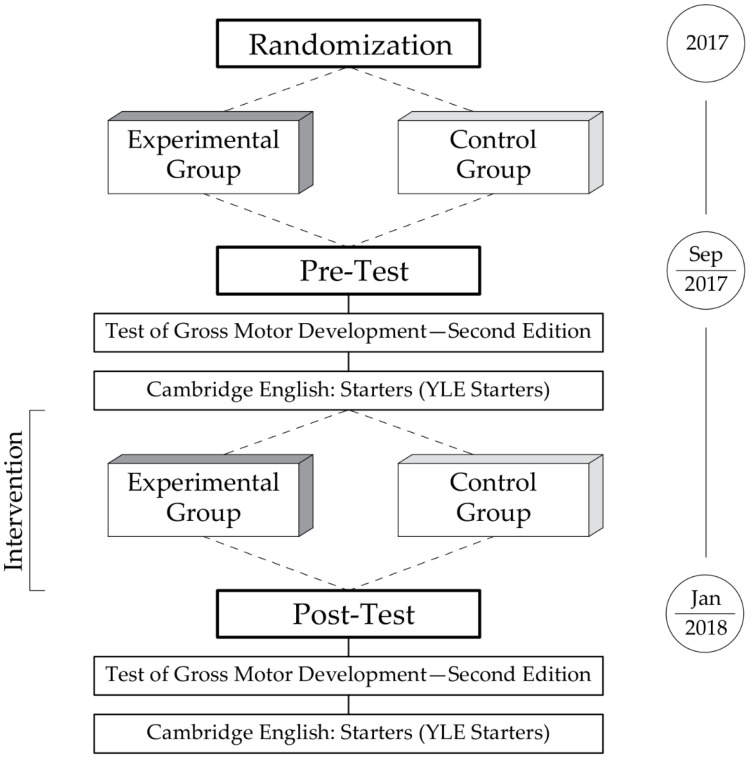
The experimental workflow. Classes were randomly assigned into experimental and control groups. Pre-test and post-test ware carried out in the same order: (1) Test of Gross Motor Development (Edition #2) and (2) Cambridge English: Starters (Pre A1 Starters).

**Figure 3 ijerph-19-08192-f003:**
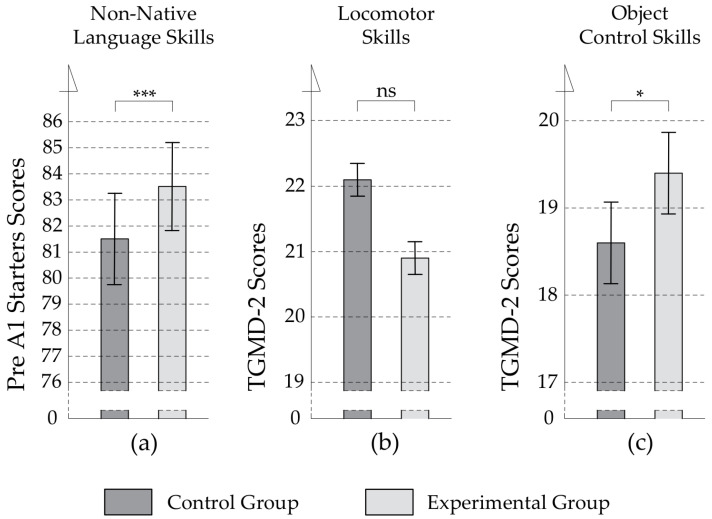
Results of the experiment in terms of non-native language and motor skills divided into two groups (non-Eduball vs. Eduball) and expressed as estimated marginal means. (**a**) The results of the two groups non-native language skills (%); (**b**) the results of the two groups in total locomotor skills; (**c**) the results of the two groups in total object control skills. Asterisks indicate significant *p*-values: * *p* < 0.05, *** *p* < 0.001 (ns—not significant). Error bars depict standard errors of the means.

**Table 1 ijerph-19-08192-t001:** Differences between groups before the intervention in terms of the non-native language and motor skills.

Skills	Control Group		Experimental Group		Mean Difference	*F*	*p*
	Mean	*SD*	Mean	*SD*			
N-N Language	63.90	18.39	68.73	12.88	4.83	0.58	0.455
Locomotor	23.00	1.13	21.86	2.96	1.14	1.79	0.199
Object Control	22.17	2.13	18.50	3.18	3.67	3.30	0.002 **

N-N—non-native. *SD*—standard deviation. *F*—*ANOVA F*-test value. Asterisks (**) indicate *p* < 0.01.

**Table 2 ijerph-19-08192-t002:** Mean and standard deviation of the control and experimental groups in the pre- and post-tests. Mean differences were calculated as the pre-test scores subtracted from the post-test scores; therefore, a positive result shows progress and a negative result shows regression.

**Skills**	**Control Group**							
	**Pre-Test**		**Post-Test**		**Mean Difference**	** *t* **	** *p* **	** *d* **
	**Mean**	** *SD* **	**Mean**	** *SD* **				
N-N Language	63.90	18.39	79.81	18.07	15.91	−5.09	<0.001 ***	>0.8
Locomotor	23.00	1.13	21.92	1.38	−1.08	2.00	0.071	0.5–0.8
*Run*	3.75	0.45	3.50	0.91	−0.25	0.82	0.429	0.2–0.5
*Gallop*	3.75	0.45	3.42	0.52	−0.33	1.77	0.104	0.5–0.8
*Hop*	4.67	0.65	4.75	0.62	0.08	−0.36	0.723	<0.2
*Leap*	2.92	0.29	2.67	0.65	−0.25	1.15	0.275	0.2–0.5
*Jump*	4.00	0.00	3.83	0.39	−0.17	1.48	0.166	0.2–0.5
*Slide*	3.92	0.29	3.75	0.45	−0.17	1.48	0.166	0.2–0.5
Object Control	22.17	2.13	19.58	2.88	−2.58	3.30	0.007 **	>0.8
*Strike*	4.58	0.67	3.50	1.00	−1.08	3.03	0.012 *	>0.8
*Dribble*	3.92	0.29	3.33	0.99	−0.58	1.87	0.089	0.5–0.8
*Catch*	2.75	0.62	2.75	0.62	0.00	0.00	1.000	<0.2
*Kick*	3.75	0.45	3.67	0.49	−0.08	0.43	0.674	<0.2
*Throw*	3.58	0.79	2.83	1.27	−0.75	1.83	0.095	0.5–0.8
*Roll*	3.58	0.67	3.50	0.67	−0.08	0.29	0.777	< 0.2
**Skills**	**Experimental Group**							
	**Pre-Test**		**Post-Test**		**Mean Difference**	** *t* **	** *p* **	** *d* **
	**Mean**	** *SD* **	**Mean**	** *SD* **				
N-N Language	68.73	12.88	84.99	12.34	16.26	−4.39	<0.001 ***	>0.8
Locomotor	21.86	2.96	21.07	1.94	−0.79	0.71	0.490	<0.2
*Run*	3.79	0.58	3.50	0.76	−0.29	1.00	0.336	0.2–0.5
*Gallop*	3.57	0.76	3.50	0.65	−0.07	0.22	0.828	<0.2
*Hop*	4.29	1.14	4.43	0.65	0.14	−0.34	0.738	<0.2
*Leap*	2.71	0.47	2.93	0.27	0.21	−1.39	0.189	0.2–0.5
*Jump*	3.71	0.61	3.43	0.76	−0.29	1.00	0.336	0.2–0.5
*Slide*	3.79	0.43	3.29	0.83	−0.50	1.84	0.089	0.2–0.5
Object Control	18.50	3.18	18.50	3.28	0.00	0.00	1.000	<0.2
*Strike*	4.00	1.04	3.71	1.44	−0.29	0.74	0.470	0.2–0.5
*Dribble*	3.29	0.91	2.29	1.14	−1.00	3.61	0.003 **	>0.8
*Catch*	2.57	0.65	2.86	0.36	0.29	−1.30	0.218	0.2–0.5
*Kick*	3.00	0.96	3.57	0.76	0.57	−2.10	0.055	0.5–0.8
*Throw*	2.57	1.16	2.57	1.34	0.00	0.00	1.000	<0.2
*Roll*	3.07	0.92	3.50	0.65	0.43	−1.31	0.212	0.2–0.5

N-N—non-native. *SD*—standard deviation. Asterisks indicate significant *p*-values: * *p* < 0.05, ** *p* < 0.01, *** *p* < 0.001. *d* < 0.2—very small or no effect, 0.2–0.5—small effect, 0.5–0.8—medium effect, >0.8—large effect [51].

**Table 3 ijerph-19-08192-t003:** Analysis of covariance (*ANCOVA*) for the non-native language and motor skills by group condition (control vs. experimental group). The result of the pre-test was set as the covariate.

Skills	*SS*	*MS*	*F*	*p*	*η_p_* ^2^
Non-Native Language	2496.00	2496.00	18.67	<0.001 ***	0.45
Locomotor	7.95	7.95	2.95	0.099	0.11
*Run*	0.61	0.61	0.88	0.359	0.04
*Gallop*	0.78	0.78	2.35	0.139	0.09
*Hop*	0.64	0.64	1.63	0.214	0.07
*Leap*	0.10	0.10	0.43	0.521	0.02
*Jump*	0.34	0.34	0.89	0.354	0.04
*Slide*	0.05	0.05	0.10	0.757	0.00
Object Control	48.08	48.08	6.07	0.022 *	0.21
*Strike*	2.23	2.23	1.44	0.242	0.06
*Dribble*	3.25	3.25	3.08	0.092	0.12
*Catch*	0.02	0.02	0.06	0.806	0.00
*Kick*	0.63	0.63	1.54	0.228	0.06
*Throw*	7.59	7.59	5.21	0.032 *	0.19
*Roll*	0.25	0.25	0.57	0.459	0.02

*SS*—sum of squares. *MS*—mean square. *F*—*ANCOVA F*-test value. Asterisks indicate significant *p*-values: * *p* < 0.05, *** *p* < 0.001.

## Data Availability

The data presented in this study are openly available in the Open Science Framework at https://osf.io/q8scy/ (accessed on 8 June 2022).

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
