# Peer review of "Physical Education with Eduball Stimulates Non-Native Language Learning in Primary School Students"

_ijerph, 2022, doi:10.3390/ijerph19138192_

Round 1

Reviewer 1 Report

Dear authors,

Concerning your article Physical Education with Eduball Stimulates Non-Native Language Learning in Primary School Students I have few suggestions:

1. At Introduction, lines 85-92 are some conclusions and it is better to move to the Conclusion.

2. At the end of the Introduction you must present the aim of the study.

3. At 2.1 present inclusion and exclusion criteria of the participants.

4. In text, lines 100-104  you mentioned that there are no significant differences between groups, but table 1 present intragroup analysis pre and post-test. I recommend to prepare another table with intergroup analysis at pre-test to support that you mentioned in text.

5. Table 3 - describe at the end of the table the abbreviation used.

6. Conclusion must be extended. Present the limitations of your study and some ideas for future studies.

Author Response

Dear Reviewer,

Sincerely,
Ireneusz Cichy
& Michal Klichowski

Reviewer 2 Report

This paper studies effects of “Eduball”, a product aiming to improve both physical and intellectual abilities at the same time, on the development of second language skills of children aged around eight. After an experiment lasting for a semester and statistical analyses, the authors find that students who attended PE classes in which Eduballs were used developed English language skills more that those who attended traditional PE classes. On the other hand, there was no difference between the two groups on locomotor skills. In my view, the results are clear-cut, and well presented in the paper. The paper can be accepted for publication with slight modifications according to the following two minor comments:

Line 87: check the spell of “improv”

Please describe what “SS” and “SM” (or “MS”?) are in Table 2.

Author Response

(The authors gave the same response as above.)

Reviewer 3 Report

My congratulations to the authors. This article is very well written and the research is interesting. But before publishing you have to attend to a couple of questions:

In the introduction it would be relevant to know how learning English is in Poland, if the media and the drawings or series are in the original version (English) or not. Because there are European countries where children grow up with series, drawings in English. It would be interesting to add a paragraph contextualizing this factor.

Material and methods: I think everything is in order and well explained, although showing table 1 before explaining the TGMD generates confusion.

Results

Table 2: What is SM, SS and F? Must be indicated in the legend of the table

Conclusion and discussion

As the authors comment, Eduball improves non-native language learning and does not affect physical performance, but different effects are shown in the control vs. experimental group. I therefore suggest a slightly broader discussion on whether EDUBALL should be introduced in the lessons of the other subjects or in Physical Education, since the students are at a sensitive stage in the acquisition of fundamental motor skills. Do we harm Physical Education in some way by learning other subjects? Would it be interesting to include part of this procedure in other subjects?

Author Response

(The authors gave the same response as above.)

Reviewer 4 Report

GENERAL COMMENTS

This is a manuscript that aimed to evaluate the effects of technique of Physical Education with Eduball for learning second language in Primary School students. This study is interesting and innovative in the Education context. It would be useful to implement new educational strategies in these levels and type of learning. However, the manuscript contains important issues that make impossible to replicate study, apart from other relevant aspects that should be considered regarding methodology like sample size justification or randomization process.

In terms of presentation, some headings need to be modified. As a crucial aspect, authors did not state the purpose of the study. Also, Introduction need to state the justification of study in a clearer way, and most of the information included refers to authors' own studies. Some information added in Methods section belongs to Results. Figures and tables should be understanding by itself.

ABSTRACT

This section lacks of important components like purpose or more details about methodology (participants, etc).

INTRODUCTION

In general, this section should context reader about the type of intervention included and the gap in the scientific literature, to end up this section with the purpose and hypothesis. Normally, information is general, without adding specific studies or details about previous interventions.

Introduction needs to add information from the total scientific body, not only self-studies. authors should explore other authors or at least, state the lack of investigation with “eduball” by other research groups.

Also, In Introduction section, it is not appropriate to detail studies, but refers to general characteristics to reach scientific gap.

Writing in Introduction should be improved in order to delete run-on sentences and make this section easier to follow.

To justify and support Eduball use, Authors state “Our studies to date [2228] show that participating in the Eduball…”. It is recommended not to support the information only with your own studies, but also with those completed in the scientific literature to avoid bias.

Line 60: can use Eduball effectively (for what?)

Additionally, one of our past experiments [27] 59 confirms that Eduball can be successfully used as therapeutic support for 60 low-performing students, particularly children diagnosed with such complex language 61 disorders as dyslexia. (to achieve what?)

Lines 69-72: run on sentence

What is the purpose of this study? What is the hypothesis? (authors state that this study “confirm this hypothesis” but, in Introduction, authors should propose the purpose and hypothesis; in latter sections, you will confirm it or not)

Last paragraph is confusing, is it referring to the present study? In that case, this information does not belong to the last part of Introduction. It should be removed.

METHODS

How randomization process was completed? Please, detail it.

How sample size was calculated? How your sample size is justified?

Lines 100-105: this information belongs to Results section, please remove from Methods.

What variables were considered for describing participants?

Table 1 refers to Results. This should move to that section.

Explanation of eduball should be included as part of the text in Methods instead of in the figure legend. Now, it is difficult to understand your experimental procedures. It would be easier to understand your intervention.

Can authors to add information about “standard PE program”?

RESULTS and DISCUSSION

Authors should avoid to interpret data in this section. It should be included in Discussion section.

Figures: it should be understanding by itself: Now, it lack of explanation of all abbreviations.

Table 1 and Table 2: it should be understanding by itself: Now, it lack of explanation of all abbreviations, units of every variable, etc.

It is recommended to add information about PE implementation in class and benefits of this method against those traditionally used (that were used as control group).

Authors should explicitly include limitations of this study.

It is recommended to add practical application of your study.

CONCLUSION

Please, be concise and state those conclusions directly derived from the present study.

Author Response

(The authors gave the same response as above.)

Round 2

Reviewer 4 Report

This is a revised paper about the effectiveness of an educational intervention based on Eduball Physical Education in second language learning of children. I congratulate the authors, they have made efforts to include a lot of revisions based on the reviewers' comments. These changes have greatly improved the quality of the report. I feel that the manuscript provides good evidence for the use of this tool in educational contexts.